# Contour Integration Underlies Human-Like Vision

Ben Lonnqvist [1]   Elsa Scialom [* 1]   Abdulkadir Gokce [* 1]   Zehra Merchant [1]   Michael H. Herzog [1]
Martin Schrimpf [1]

## Abstract

Despite the tremendous success of deep learning in computer vision, models still fall behind humans in generalizing to new input distributions. Existing benchmarks do not investigate the specific failure points of models by analyzing performance under many controlled conditions. Our study systematically dissects where and why models struggle with contour integration – a hallmark of human vision – by designing an experiment that tests object recognition under various levels of object fragmentation. Humans (n=50) perform at high accuracy, even with few object contours present. This is in contrast to models which exhibit substantially lower sensitivity to increasing object contours, with most of the over 1,000 models we tested barely performing above chance. Only at very large scales ($\sim 5B$ training dataset size) do models begin to approach human performance. Importantly, humans exhibit an integration bias – a preference towards recognizing objects made up of directional fragments over directionless fragments. We find that not only do models that share this property perform better at our task, but that this bias also increases with model training dataset size, and training models to exhibit contour integration leads to high shape bias. Taken together, our results suggest that contour integration is a hallmark of object vision that underlies object recognition performance, and may be a mechanism learned from data at scale.

## 1. Introduction

Picture a simple drawing of a pan where most of its contours are erased, leaving only a set of disconnected frag-

*Equal second authors [1]École Polytechnique Fédérale de Lausanne (EPFL), Switzerland. Correspondence to: Ben Lonnqvist <benhlonnqvist@gmail.com>.

*Proceedings of the 42nd International Conference on Machine Learning*, Vancouver, Canada. PMLR 267, 2025. Copyright 2025 by the author(s).

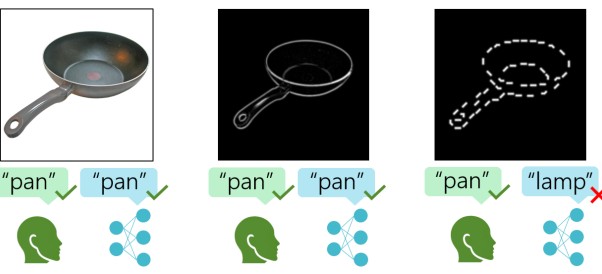

*Figure 1.* Human and DNN categorization of (*left*): a standard RGB image. (*Middle*): a contour-extracted image. (*Right*): a fragmented image requiring contour integration. The vast majority of over 1,000 tested models catastrophically fail at a categorization task the moment object contours are fragmented.

ments. You would most likely have no trouble recognizing the drawing as a pan – after all, your visual systems "fills in the blanks" and allows you to see the pan regardless of its fragmentation (Figure 1). This is an example of contour integration, a well-established mechanism in the primate brain, whereby disconnected elements are *integrated* into a continuous form, enabling recognition. Contour integration is highly relevant to naturalistic vision due to the number of occlusions in the natural environment, and it yet remains unclear as to why the human brain is so robust to this particular type of image perturbation.

While deep neural networks (DNNs) reach superhuman performance at standard image recognition benchmarks like ImageNet (Russakovsky et al., 2015), their generalization capabilities to unseen image distribution shifts lags behind human generalization capabilities (Geirhos et al., 2021; Baker et al., 2018; Bowers et al., 2023; Biscione & Bowers, 2023; Malhotra et al., 2022; Muttenthaler et al., 2023; Malhotra et al., 2023). Despite this large body of evidence suggesting differences between human and model generalization capabilities, the causes behind the differences remain unclear.

A key reason for the difficulty in understanding model failures is a lack of systematic studies that investigate individual well-established mechanisms in the primate brain. By lack of systematicity we mean that studies typically fall short on one of two axes: first, they may not investigate a set

of related experimental conditions that would allow them to know where exactly models diverge from human behavior. Second, studies simply may not include a sufficiently large model set to allow a quantitative explanation of model behavior at scale. For example, simply knowing whether models integrate contours would offer limited insight; knowing *how* to improve them is crucial. In this work, we seek to bridge this explanation gap using contour integration as an exemplary test case.

In short, we develop a new contour integration task that we tested 50 humans and over 1,000 DNN models on. Importantly, our task consists of a grand total of 20 different conditions: two baseline conditions with full color (RGB) images, and contour-filtered images; as well as two different sets of fragmented contour images with 9 sets of levels of fragmentation in each. Using this comprehensive experiment alongside our gigantic model set, we make several contributions:

1. We found that most models struggle on our task that is relatively easy for humans, especially in more challenging settings. For example, humans still achieved 50% accuracy with only 35% of fragmented elements remaining, while models were barely above chance even with decoder adaptation.

2. Models that were larger and trained on more data performed better at our task. For instance, GPT-4o's performance was near-indistinguishable from humans; but this trend held across all models.

3. We found that models that are trained on more data had a larger integration bias (performance on directional segment stimuli minus performance on directionless phosphene stimuli), suggesting that contour integration is critical for human-like object recognition performance – indeed, a larger integration bias translated into higher accuracy as well as better robustness.

4. We trained models on contour integration, and found that they not only acquired an integration bias, but also a shape bias greater than that of shape-trained models.

5. Together, these suggest that contour integration underlies human-like vision, and may be a mechanism learned from large-scale data, rather than an inbuilt mechanism inherent to biological vision.

## 2. Related work

**Studies of visual behavior in DNNs.** Most work studying visual behavior in DNNs has focused on a number of out-of-distribution datasets examining different cases of human-model alignment due to DNNs' tendency to exploit statistical regularities in training data in naturalistic settings (Beery et al., 2018; Geirhos et al., 2020a). Geirhos et al. (2018) and Baker et al. (2018) tested several DNNs on a shape bias task and found that DNNs make categorization decisions largely based on image texture content, rather than shape content. This work was expanded later with a larger set of experimental data spanning different data distribution shifts (Geirhos et al., 2021). A key finding of the work was the observation that amongst a set of 52 models, the models trained with the most data appeared to make the most human-like categorization errors, while models trained on less data made more idiosyncratic errors. (Hermann et al., 2020) argued that data augmentations can also help lead to less texture bias.

Gestalt grouping (Wagemans et al., 2012; Biederman, 1987), the ability to combine object parts into wholes, has also seen a rise in popularity, though experimental results are mixed. While some report that DNNs fail to capture principles of Gestalt grouping (Bowers et al., 2023; Malhotra et al., 2023), others report mixed or partial success (Biscione & Bowers, 2023; Pang et al., 2021; Lonnqvist et al., 2024). In the more naturalistic domain, Muttenthaler et al. (2023) studied model behavior using an odd-one-out task, finding model objective function and training dataset size to drive performance, while model scale and architecture had no effect.

**Contour integration in humans and primates.** Contour integration in general has been studied behaviorally extensively in humans. There are several well-established hallmarks of contour integration, most important for us is the effect of *contour alignment* – when individual fragments align, human performance substantially increases and detection becomes possible (Polat & Sagi (1994); Kovács & Julesz (1993); Roelfsema (2006); Elder & Zucker (1993); see also Wagemans et al. (2012); Biederman (1987)). Contour integration has also been studied extensively in animal models using electrode arrays to make direct neural recordings of populations of neurons. Of particular note is primary visual cortex V1, where neurons respond strongly to arrangements of aligned fragments (Li et al., 2006; Kapadia et al., 1995; Bosking et al., 2000; 1997). In addition, evidence of contour integration has also been found in visual area V4 (Chen et al., 2014).

**Contour integration in DNNs.** While some of the earliest non-DNN models of contour integration were studied earlier (Li, 1998), classical contour integration was studied in DNNs by Linsley et al. (2018), who showed that traditional convolutional neural networks struggle to learn to integrate line segments across the visual field in an abstract task. Inspired by horizontal connectivity in the primate brain, they developed a horizontal gated recurrent unit network, which was able to learn and solve the contour inte-

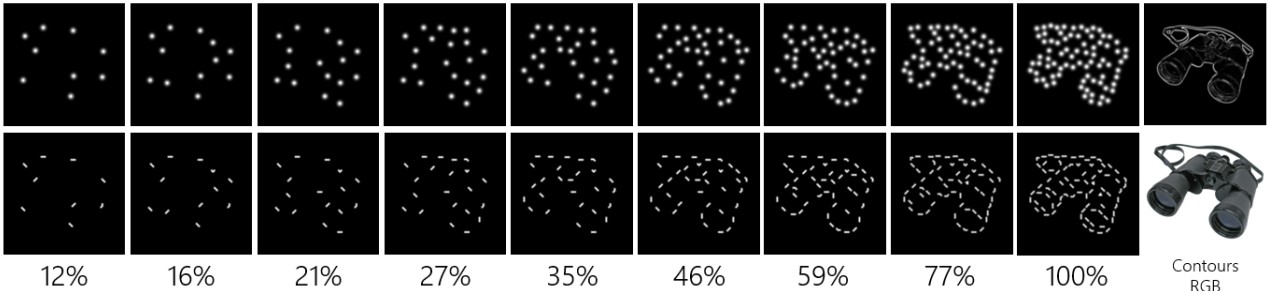

Figure 2. **Stimulus examples.** Numbers refer to the number of fragments in the stimulus relative to the 100% maximum. 100% is the maximum number of fragments placed along the object contours without overlap. *Top*: Phosphene stimuli ranging from 12% to 100% number of fragments. *Bottom*: Segment stimuli. *Right*: Contour and RGB stimuli.

gration task. (Funke et al., 2021) took a different approach: they fine-tuned an ImageNet-trained ResNet-50 (He et al., 2016) on line segment images with open and closed contours. They found that while the model did well on the task in-distribution, generalization proved catastrophically difficult.

Our work differs from previous work on several critical axes. First, we design a dataset that allows for a more detailed examination of model and human behavior than previous datasets, such as shape bias datasets (Geirhos et al., 2018). This is due to the fact that we collected data to stimuli across many different conditions, and many different levels of fragmentation in our stimuli. Second, the scale of our model evaluation eclipses previous studies by an order of magnitude and enables us to quantitatively establish links that were not possible prior. Third, the combination of the dataset design and large model set allow us to analyze our data in novel ways. We are able to quantitatively show that contour integration leads to improved object recognition, and better robustness to image perturbations.

## 3. Methodology

**Human psychophysical experiment.** We recruited 50 human participants to take part in a 12-alternative forced choice object recognition task. All participants performed the task in a controlled laboratory setting, with stimuli presented on a high-quality 1920x1080 pixel monitor. Upon arrival, the participants were sat in a darkened room where they were first verified for normal or corrected-to-normal vision using the Freiburg acuity test (Bach, 1996). Participants also performed a short language test with unrelated object categories, as well as a short set of 24 familiarization trials with object categories not included in our main experiment, intended to familiarize themselves with the response options and trial structure of the task. In the main experiments, participants were divided into two groups of 25 participants, with each group being presented either phosphene stimuli (Figure 2 *top*) or segment stimuli (Figure 2 *bottom*), as well

as both the contour and full color (RGB) conditions. Stimuli were presented in ascending order of fragmentation starting from 12%, up to 100% followed by contours and RGB. All stimuli were presented foveally spanning $8 \times 8$ degrees of visual angle. Stimuli were presented for 200ms, followed by a 200ms $1/f$ noise mask. After the noise mask, participants could select their response option from 12 categories arranged in a circle around the center of the screen. In total, we report results for 50 participants, totalling 26,400 trials. For more details on experimental procedure and stimulus selection, see Appendix A.2. Additionally, 10 new participants performed an additional control experiment to control for learning effects due to the ascending stimulus presentation order (details in Appendix A.3).

**Stimulus synthesis.** For our RGB base images we used the BOSS dataset (Brodeur et al., 2010; 2014) which consists of high-quality background-extracted images of everyday objects. We generated a total of 19 different datasets (Rotermund et al., 2024) from these images: contour-extracted images, as well as nine different levels of fragmentation for each of our two experimental conditions (directionless phosphenes and directional segments). The levels of fragmentation ranged from 100% (meaning the maximum number of elements we have in any image), down to 12% of that maximum following a log scale. We used 4 objects per category in our experiment for each of our 12 object categories, resulting in 432 fragmented stimuli for phosphenes and segments respectively. Combined with the RGB and contour stimuli, this resulted in a total of 528 stimuli per human participant. At the time of submission, these stimuli are not public and could therefore not have been used for model pre-training.

**Model responses.** To investigate whether models could do our task out-of-the-box, as well as whether their representations could support our task, we used Brain-Score (Schrimpf et al., 2020) to extract model responses in two different ways:

Zero-shot: To directly test DNNs' capability to perform the

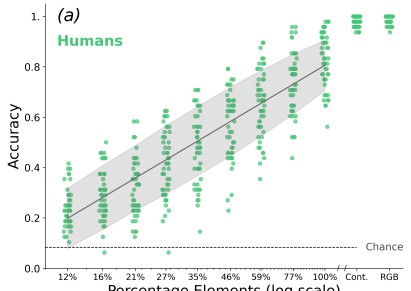 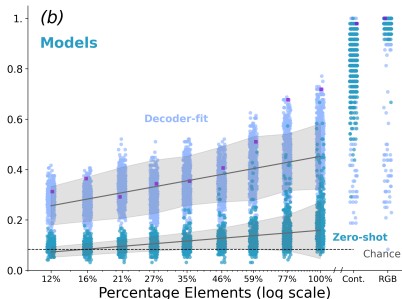 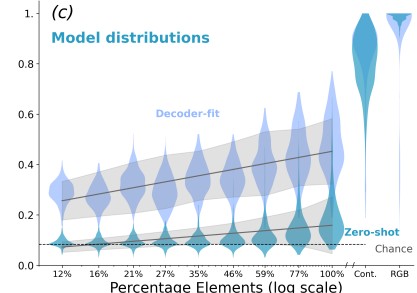

*Figure 3.* **Human and model performance.** Individual dots represent individual **participants'** or **models'** average performance, and error bars show 95% confidence intervals. Task chance performance is 8.33%. ***(a)***: Human object recognition performance. The x-axis shows different conditions (Figure 2), combined across phosphenes and segments. ***(b)***: Model categorization performances. **Dark blue**: Zero-shot models with responses mapped from ImageNet labels *without* additional fitting. **Light blue**: Linear decoder-fit models with an additional 120 supervised trials for a linear decoder within-condition. **Violet**: GPT-4o zero-shot performance. *(c)*: The same data as in *(b)*, visualized as distributions.

fragmented object recognition task in a human-comparable manner, we mapped model responses from ImageNet categories (Russakovsky et al., 2015) to our 12 object categories zero-shot using a WordNet synset mapping (Geirhos et al., 2021). This was only possible for models that output ImageNet 1000-class labels. We also tested **GPT-4o** zero-shot, for which you can find details in Appendix A.12.

**Decoder-fit**: For each of our 12 object categories, we first selected 10 ImageNet images from the corresponding ImageNet categories. We then removed backgrounds from these images (Gatis, 2023) and generated 120 novel fragmented images for all percentage levels; 10 images per object category. We fit linear decoders on the penultimate layer activations.

**Model selection.** Our model set consisted of several sources of models: 505 pre-trained models from the timm library (Wightman, 2019) and 23 pre-trained models from the taskonomy library (Zamir et al., 2018). In addition, we also added 514 models trained ourselves (see Appendix A.7 for details). These additional models ranged from very small networks trained on a few hundred samples, to medium-sized models trained on ImageNet-21k. The largest pre-trained models from the timm library were trained on over 5 billion images. In total, we include 1,038 models from 13 architecture families and 18 datasets in our work, including Vision Transformers (Dosovitskiy et al., 2021), Convolutional Neural Networks (Liu et al., 2022), and many others, trained on datasets such as LAION-2B (Schuhmann et al., 2022), CLIP (Radford et al., 2021), and ImageNet (Russakovsky et al., 2015). For a quick list of models analyzed in this study, as well as their architectures, see Table 1. For more details about the models, including information about their training datasets and compute, see Appendix A.7.

**Model and human comparison.** We computed model and human accuracy across conditions and perform appropri-

ate statistical tests throughout. There are several reasons for this choice. In recent years there has been a trend to move away from accuracy towards fine-grained error-based metrics (like error consistency, Geirhos et al. (2020b)). A key motivation for this shift has been that it is increasingly difficult to find benchmarks on which humans outperform leading deep learning models. While these metrics are extraordinarily valuable, they also come with downsides such as a large amount of noise in fitting estimation that requires up to hundreds of trials per condition in a standard experimental setup. As such, analysis at the level of conditions, rather than stimuli, is unintuitively more difficult using these metrics. Because the goal of our study is to explain the entire range of behaviors across a set of conditions, we chose to rather increase the difficulty of our task to avoid ceiling effects, and estimate performance simply using accuracy.

*Table 1.* Number of models and subjects in the main experiments.

| Architecture | *N* **Zero-shot** | *N* **Decoder-fit** |
| --- | --- | --- |
| ViT | 122 | 257 |
| ConvNeXt | 107 | 189 |
| ResNet | 75 | 179 |
| EfficientNet | 45 | 87 |
| ResNeXt | 37 | 39 |
| MaxViT | 26 | 30 |
| SwinTransformer | 26 | 34 |
| MobileViT | 16 | 16 |
| CORnet-S | 15 | 30 |
| AlexNet | 15 | 30 |
| FastViT | 14 | 14 |
| RegNet | 13 | 16 |
| TinyViT | 8 | 11 |
| **Total** | **621** | **1038** |
| **Humans** | **50**+10 control | |

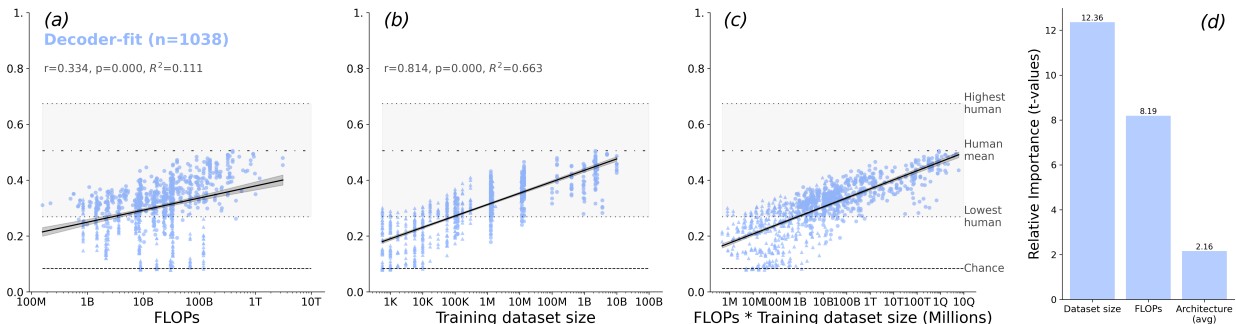

*Figure 4.* **Dataset scale and model compute explain performance.** *(a) and (b)*: Model performance on fragmented objects is explained moderately well by model compute measured in total floating point operations (FLOPs) per sample ($r = 0.334$, $p < 0.001$, and very well by training dataset size ($r = 0.814$, $p < 0.001$). *(c)*: Considering both variables together (visualized by multiplying them) results in a smooth trend. *(d)*: Dataset size and model compute per sample are more important for explaining model results than architecture. *y-axis* shows t-values from a multiple regression. To get the average effect of architecture, we averaged the absolute values of architecture t-values.

## 4. Results

**Humans substantially outperform models.** We first analyzed human performance and compared it to the performance of a set of 528 pre-trained models from the timm and taskonomy libraries (Figure 3). Humans performed at ceiling accuracy on the contour and RGB conditions, similarly to most models. This is in contrast to our experimental conditions where we fragmented the object contours – human performance scaled approximately log-linearly (Figure 3*(a)*; $accuracy = 0.29 * \log(x) - 0.51$, $R^2 = 0.73$, $p < 0.001$) in the number of elements. While model performance followed a similar scaling function in the number of fragmented elements, performance scaled substantially slower than in humans, especially zero-shot (Figure 3*(b,c)*; **zero-shot** $accuracy = 0.04 * log(x) - 0.03$, $R^2 = 0.23$, $p < 0.001$; **decoder-fit** $accuracy = 0.09 * log(x) + 0.03$, $R^2 = 0.39$, $p < 0.001$). Of particular note are zero-shot models, which catastrophically failed on our task, with only few models reaching above-chance performance in general. However, **GPT-4o** stood as an outlier in zero-shot models, reaching near human levels of performance and being the best zero-shot model by a large margin.

Decoder-fit models performed substantially above zero-shot models, with both a larger slope and intercept than zero-shot models. However, while decoder-fit models' performance started at approximately the human level on the most difficult condition (12% percentage elements), their performance on average did not scale to the human level at the easiest fragmented condition (100% percentage elements). Although the overall distribution of pre-trained models did not align with the human distribution of results, there was individual variation in model results, with a small number of models substantially outperforming others.

**Dataset size and model compute explain differences in model performance.** To understand the individual differences between the models, we studied whether model performance could be explained by three simple factors: model training dataset size, model compute per sample, and model architecture (Figure 4). In addition to the 528 pre-trained models in our selection, we also trained an additional 514 models primarily on the lower end of the training dataset size and model compute axes to investigate the full range of scaling performance across pre-existing as well-controlled training diets. Since model performance across the board is substantially higher for decoder-fit models, we focus the rest of our analysis on these models. For full details on models and training diets, see Appendix A.7.

Across our entire set of 1,038 models, we find that both model compute per sample (reported as floating point operations or FLOPs per input image), as well as the training dataset size (reported as the number of unique images in the training dataset of the model) are both important factors explaining model performance (Figure 4*(a,b)*). Training dataset size is especially important – regressing training dataset size on model results on our task performance results in a correlation of $r = 0.814$, explaining most of the variance in the data. Due to the high multicollinearity between FLOPs and training dataset size, we also show a simple multiplication of the two for visualization purposes (Figure 4*(c)*).

**The importance of architecture for performance.** We considered the relative importance of model compute per sample, model training dataset size, as well as the average effect of architecture (Figure 4*(d)*), as t-values of a multiple regression. We averaged the absolute values of the t-values of the effects of architecture families on model performance, and observe that architecture on average is

relatively unimportant for contour integration performance on our task compared to model compute or training dataset size.

**Directional segments reveal contour integration in humans and models.** To analyze the degree of contour integration caused by the effect of adding directional segments (Figure 2, *bottom*) instead of directionless phosphenes (Figure 2, *top*), we compared human and model performance on these two conditions. Results are shown in Figure 5.

Humans exhibited a clear preference for directional segments over directionless phosphenes, with a large effect size (Cohen's $d = -1.96$). Given the overall poor average performance of models in the task in both the zero-shot and decoder-fit regimes compared to human performance, we were somewhat surprised to find that both types of model responses exhibited a human-like preference for directional segments, with statistically significant effect sizes (**Cohen's** $d = -0.84$; **Cohen's** $d = -1.55$ respectively). This suggests that while models were not as performant on our contour integration task as humans, they share a similarity in how they solve the task.

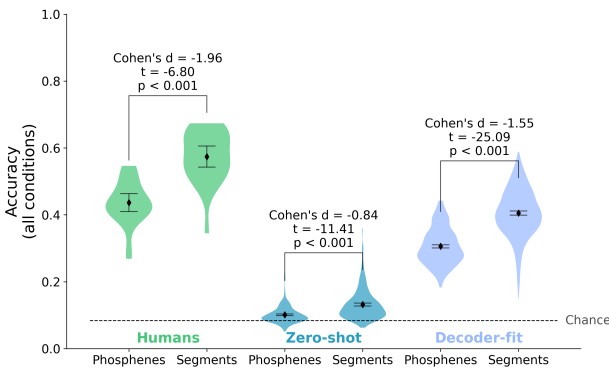

*Figure 5.* **Human and model performance on phosphenes and segments.** Humans exhibited higher performance when presented with segments, rather than phosphenes. Dots show human and model means, and error bars are 95% confidence intervals.

**Contour integration explains human-level performance.** To understand *why* larger models trained on more data do substantially better than smaller models trained on less data, we sought to analyze what we call *integration bias*: the performance difference between the segment task and the phosphene task (Figure 6(a)). Since model integration bias was smaller than that of humans' on average (Figure 5), and because models exhibited a wide range of performance levels largely explained by their training dataset size, we conjectured that these model-to-model performance differences actually stem from the models' ability to integrate contours. In other words, if integration bias is a crucial driver of overall performance, it should correlate with model

performance across our model set.

This is exactly what we found in Figure 6(b): the models with the largest integration biases are the most human-like in performance, and model average performance increased with integration bias. Together these suggest that contour integration is crucially related to human-like object recognition regardless of the underlying architecture the computation is implemented in.

**Contour integration is learned from data.** We tested whether the size of the model's training dataset directly correlated with its integration bias across our 1,038 decoder-fit models (Figure 6(c)). The results revealed a strong positive correlation ($r = 0.594, p < 0.001$), indicating that contour integration is a mechanism models learn from the data distribution automatically, rather than a mechanism that must be hand-engineered in the architecture.

This finding carries an important implication. It has traditionally been thought that contour integration is largely a product of horizontal connectivity in primary visual cortex (Bosking et al., 1997; 2000; Kapadia et al., 1995). Here we demonstrate that not only are early horizontal connections not necessary, but that this mechanism can and *is* learned directly from a large enough training dataset.

**Contour integration leads to increased robustness.** We selected a subset of 50 models from our set of models to evaluate on robustness to image perturbations (ImageNet-C-top1). The models were selected linearly based on performance on our fragmented object task. We find that the larger the integration bias the model had, the better its performance was against image perturbations (Figure 6(d); $r = 0.671, p < 0.001$).

**Contour integration leads to shape bias.** We wanted to study whether it would be possible to directly train models to group elements, and whether that would induce both integration bias and shape bias. To that end, we trained several ResNet-18 (He et al., 2016) models on ImageNet-1k (Russakovsky et al., 2015), and several different manipulations of ImageNet-1k, training on the following data setups:

**IN only**: ImageNet-1k baseline with no additional images.

**IN + Phosphenes**: ImageNet-1k combined with grayscale images with directionless phosphenes.

**IN + Segments + Phosphenes**: ImageNet-1k combined with both segment and phosphene stimuli.

In addition, we tested a **Shape-biased** model (Geirhos et al., 2018). All of our fragmented object models were trained for 100 epochs using two A100 GPUs per model, with a total batch size of 512. For more training and data preparation details, see Appendix A.9, and for additional controls on contour training, see Appendix Figure 12.

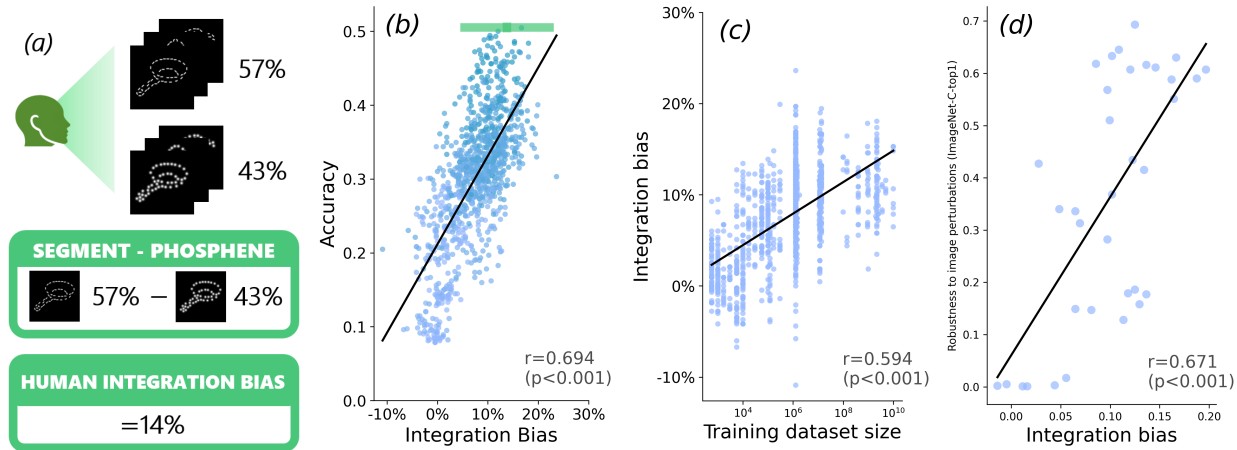

Figure 6. **Integration bias explains model performance.** *(a)*: We defined integration bias as the difference in performance between the segment condition and the phosphene condition. For example, a model achieving 60% performance on segments and 50% performance on phosphenes would have 10% integration bias. *(b)*: Light blue dots are individual decoder-fit models. Green lines are human 95% confidence intervals centered around the mean. Model integration bias correlated strongly with model performance. Darker color indicates larger model training dataset size. *(c)*: Model training dataset size correlated with integration bias. *(d)*: Model integration bias led to higher robustness as measured by ImageNet-C (top1 accuracy). Included are 50 models selected based on performance on our task (Appendix A.10).

We found that while it is possible to improve performance on fragmented objects by directly training models on similar types of stimuli, it comes at a cost – unlike previous studies (Geirhos et al., 2018), we were not able to simultaneously improve performance on full color images while also training models to exhibit improved robustness to contour fragmentation (Figure 7). Rather, models that improved performance on fragmented stimuli (**IN + Segments**, **IN + Phosphenes**, **IN + Segments + Phosphenes**) also saw minor drops in full color image recognition (RGB). While this is expected, it also paves the way to study contour integration mechanistically in smaller DNN models.

What is surprising, however, is that not only did our models acquire a shape bias (Figure 7, *right*; measured by Geirhos et al. (2021) cue conflict accuracy), but their shape bias was higher than that of a shape-bias trained ResNet-50 (Geirhos et al., 2018). Simultaneously, the integration bias acquired by the shape-biased model was barely half of the human equivalent. Taken together with evidence from neural (Roelfsema, 2006; Kapadia et al., 1995) and psychophysical (Polat & Sagi, 1994; Elder & Zucker, 1993) results, these findings point us to a conclusion: contour integration leads models to acquire a shape bias.

## 5. Discussion

**Summary**. This work presented a large-scale comparison of human and DNN performance on a contour integration task. By carefully designing an experiment meant to specifically test the failure points of DNN models, we found that across an evaluation of over 1,000 DNNs, they fail to capture human behavioral performance, even when allowing for additional decoder-fitting trials. We found that the root cause of the model failure is an inability to *integrate contours* as measured by the difference in performance on images with directional segments and directionless phosphenes: models that had a larger *integration bias* also performed better on our contour integration task in general. Our definition of integration bias builds on a vast body of evidence both in human behavior and neural data that shows the specificity of contour integration to directionally aligned line segments (Bosking et al., 2000; 1997; Roelfsema, 2006; Kovács & Julesz, 1993; Kapadia et al., 1995). We found that integration bias arises from model training on large datasets – models trained on more data exhibited a larger integration bias. Finally, training models on fragmented images led not only to integration bias, but also to a larger shape bias than shape-trained models (Geirhos et al., 2018). This allows for researchers to study the computations underlying human-like integration bias in DNNs, and to obtain a high model shape bias simultaneously with a high integration bias, by training on fragmented objects.

**Limitations.** In this work, we focused on observational findings. We have made several efforts to remedy this: first, our model set covered much of the extant landscape of open access pre-trained vision models and, combined with models we additionally trained ourselves, constituted what is the largest human-model comparison to date by approximately an order of magnitude. Second, we synthesized new perturbed versions of ImageNet and trained models on different

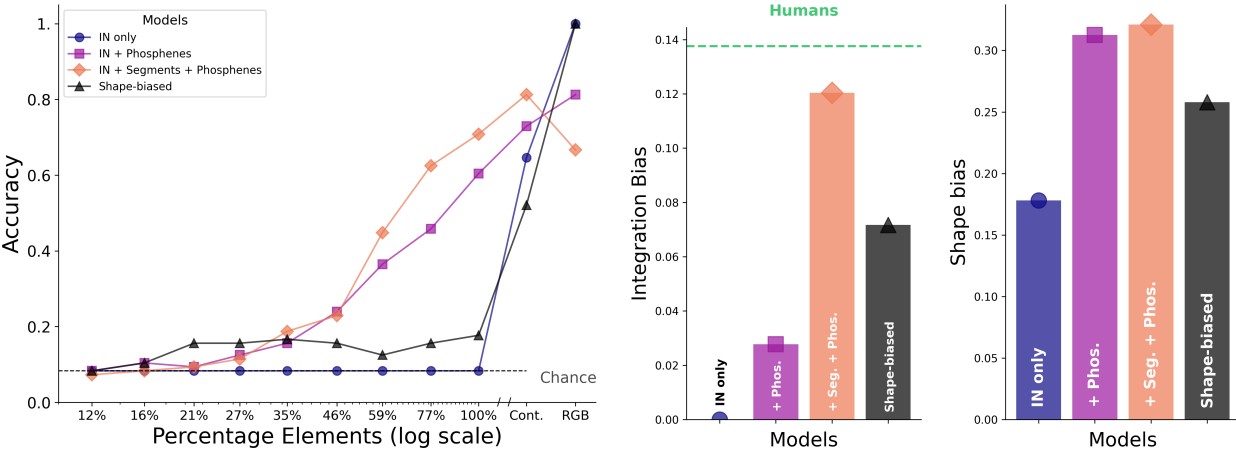

*Figure 7.* **Model performance of models trained on ImageNet-1k and fragmented ImageNets.** All models were evaluated zero-shot. *Left*: Our trained model performance on the fragmented object task. *Middle*: Individual models' integration bias. *Right*: Model shape bias as measured by the cue conflict dataset (Geirhos et al., 2021). Our models trained on phosphenes, and a combination all reach higher shape bias than the Stylized ImageNet-trained model (Geirhos et al., 2018).

combinations of ImageNet and its perturbed counterparts.

Another limitation of our study was that we did not evaluate leading vision-language models at the same scale we evaluated pure vision models. We took steps to address this: we included several CLIP-trained models in our training set, and while CLIP-trained vision encoders are not joint vision-language models, their representations are nevertheless guided by language. We also included GPT-4o in our model comparison, which we found to be the most human-like zero-shot model.

Finally, since we used pre-trained models in most of our evaluation, it is possible that some models may have been trained on images that are similar, but not identical, to the ones we used to test models. We took great care to avoid leakage to any models' training datasets: we synthesized the stimuli ourselves using a stimulus rendered (Rotermund et al., 2024), and none of the stimuli we used in the study were published at the time of model evaluation.

**Future directions.** Our study presents an interesting outlook for the field of human vision modeling. On one hand, our results may give clues as to why many efforts at modeling the edge cases of human vision, such as Gestalt grouping, have failed (Bowers et al., 2023; Baker et al., 2018; Malhotra et al., 2022). A task as simple as contour integration, believed to occur mainly in primates in primary visual cortex V1 (Kapadia et al., 1995; Li et al., 2006), requires gigantic models trained on billions of images, as well as additional decoder trials to reach human performance. Furthermore, attempts to model human behavior directly by fitting a relatively simple model with standard hyperparameters yielded mixed results, given that while we were able to

improve contour integration performance, full color image object recognition performance decreased. Nevertheless, the models exhibited greater shape bias than even those trained specifically for it, while the shape-biased model did not have as large of an integration bias as models trained for contour integration. This demonstrates that shape bias emerges in models that also achieve contour integration. Human visual behavior is astonishingly complex, and our work shows that some of that elusive complexity nevertheless emerges from large enough training datasets. In other words, our findings challenge the notion that hand-engineered features or inductive biases are necessary to replicate even some of the most fundamental mechanisms of human-like visual processing.

We believe this highlights a potential way forward for building human-like vision models. In our study we saw that models, particularly zero-shot, fail to replicate human behavior catastrophically across the board. In addition, our results show that the primary reason for this is *integration bias*, and particularly how it improves with dataset scale. This highlights an interesting possibility: perhaps the reason models fail to replicate human-like behavior is not because they are not large enough *per se*, but because their training datasets have systematic differences to human visual input (see e.g. Long et al. (2024)). One such example is the relative absence of occlusions in internet photographs – photographers often prefer taking photos with better form and figure than the visual input that humans observe on a momentary basis (Leung et al., 2021), but other biases exist too (Torralba & Efros, 2011). Instead of attempting to build better human-like models through architectural changes, our results highlight the possibility that in search of human-like models, we must seek human-like training data.

## Acknowledgements

We thank Yingtian Tang for helpful discussions, and Marc Repnow for technical support.

## Impact Statement

Our findings challenge conventional assumptions about the role of architectural design in visual tasks, highlighting instead the critical role of large-scale data and specific biases like contour integration. These insights pave the way for more human-like and robust AI vision systems.

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

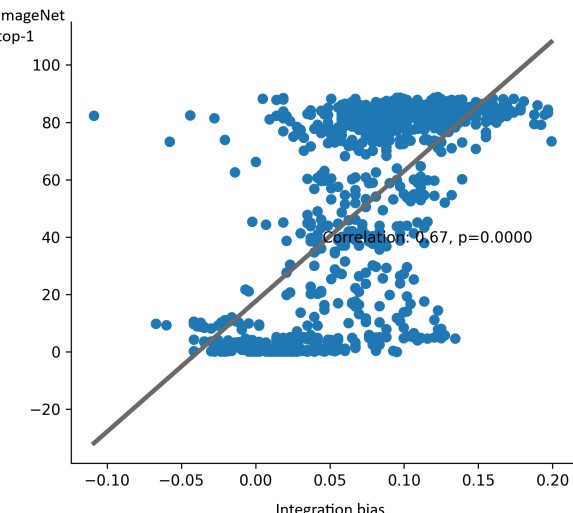

*Figure 8.* **Integration bias explains ImageNet classification performance.** Correlation between ImageNet performance (top1) and integration bias.

## A. Appendix

### A.1. Integration bias predicts classification accuracy

We evaluated model performance on ImageNet and found the same results as we did on ImageNet-C (Figure 4). Integration bias predicts performance across tasks, from fragmented stimuli (Figure 6a) to ImageNet ($r = 0.67, p < 0.01$).

### A.2. Human experiment category selection

We included 12 categories in our task: `truck`, `cup`, `bowl`, `binoculars`, `glasses`, `hat`, `pan`, `sewing machine`, `shovel`, `banana`, `boot`, `lamp`. The object category selection was done on the basis of a pilot experiment, where 46 different in-lab participants performed free-naming trials of similar objects without a strict time limit. The 12 categories tested here all reached at least 90% free-naming agreement among the pilot participants.

### A.3. Human control experiment

To control for whether our experimental results could have been influenced by any human unsupervised learning or habituation during the experiment, we ran an additional control experiment. We recruited 10 human participants (5 females; mean age $= 21.3$ years, $\sigma = 3.31$). We followed the same experimental methodology as in the main experiment, splitting the participants into two equal groups of segments and phosphenes. The only difference was that these participants only performed the condition with 100% fragments alongside the contour and RGB conditions, instead of all conditions from 12% to 100%. We found no difference in pilot participants' performance in the 100% condition compared to the main experiment, ruling out the possibility of unsupervised learning or habituation. Results are shown in Figure 9.

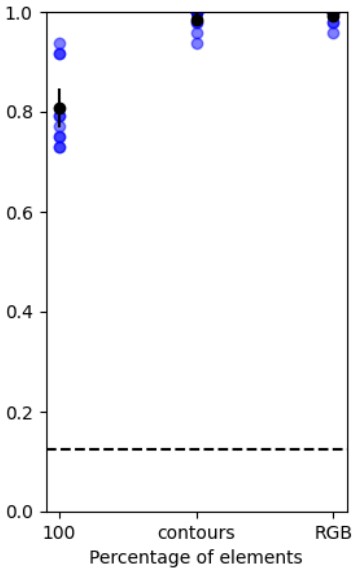

*Figure 9.* **Human control experiment.** Individual dots represent individual participants

### A.4. Human performance scaling across conditions

We analyzed the difference in how performance across the segments and the phosphenes conditions scales. We found little difference (phosphenes vs segments) in humans – the difference in performance in these two cases is mostly a shift of intercept instead of a change in slope across the number of elements (figure **??**), though while the effect of difference in slopes is small, it is still technically significant ($t = 2.87, p < 0.01$). See Figure 10 for details. This means that the directionality of the segments only slightly affects performance positively over the directionless prosphenes as element density increases.

### A.5. Integration is not predicted by receptive field size alignment

We evaluated the same subset of models as in Figure 6 on two Brain-Score benchmarks that test for the similarity of the effective receptive field size of a model to a primate counterpart. These are the Grating summation field (GSF) and surround diameter (Marques et al., 2021; Cavanaugh et al., 2002) benchmarks measured in Macaque V1. See Figure In short, we do not find a statistically significant relationship between either measure of receptive field size similarity and fragmented object recognition accuracy (GSF $r = 0.2655, p = 0.0653$; surround diameter $r = 0.276, p = 0.055$)

### A.6. Contour-trained models do not achieve as high shape bias or integration bias

As in Figure 7, we also trained additional control models to investigate whether contour-based models, and the distribution shift in image statistics caused by that, could be at cause for the results in Figure 7. We trained the following models:

**IN only**: ImageNet-1k baseline with no additional images.

**Contours only**: Contour-extracted grayscale images.

**IN + Contours**: ImageNet-1k combined with contour-extracted images.

**IN + Binary contours**: ImageNet-1k combined with a dataset where we use a $3 \times 3$ Gaussian filter with Otsu thresholding on contour-extracted images that binarizes contours in the image.

**IN + Segments**: ImageNet-1k combined with grayscale images with directional segments.

**IN + Phosphenes**: ImageNet-1k combined with grayscale images with directionless phosphenes.

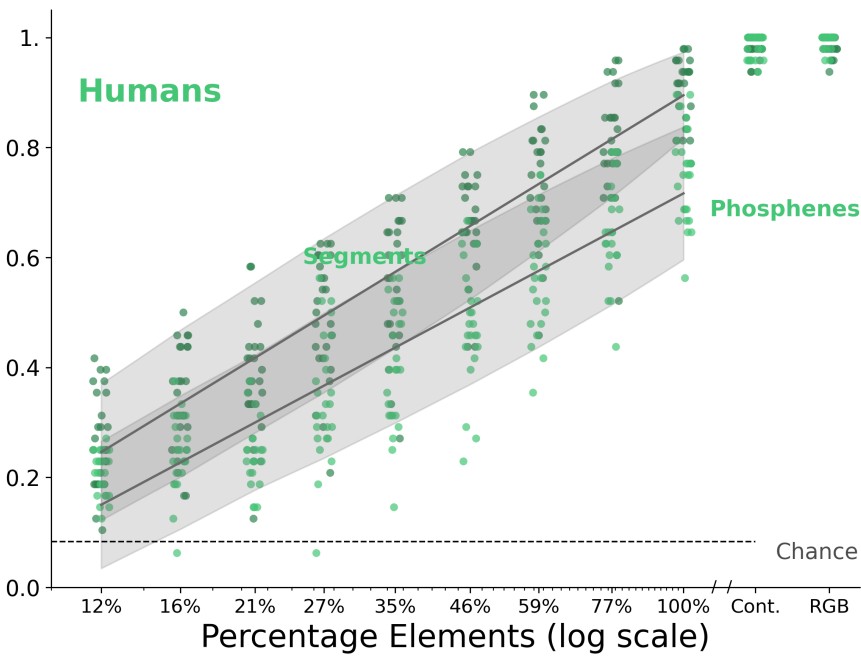

*Figure 10.* Human accuracy split by condition (phosphenes: light green; segments: dark green).

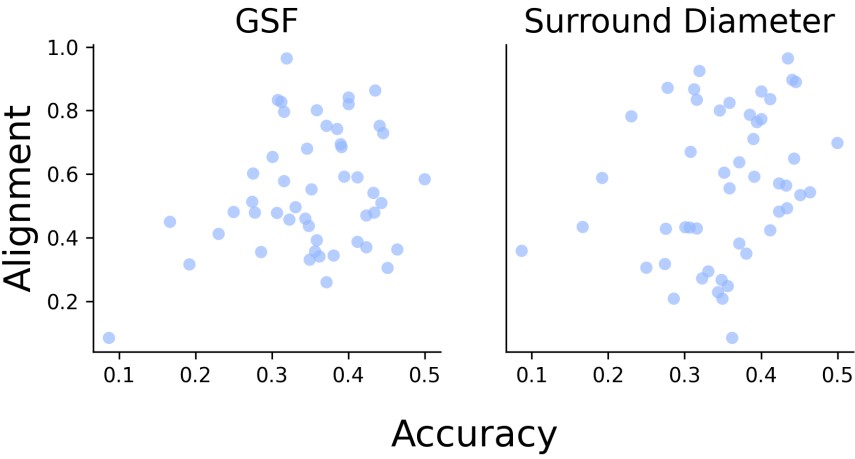

*Figure 11.* Model performance correlated to alignment in two benchmarks: the Grating summation field (GSF) and surround diameter (Marques et al., 2021; Cavanaugh et al., 2002).

**IN + Segments + Phosphenes**: ImageNet-1k combined with both segment and phosphene stimuli.

We found that the effect of training solely on contours is minimal to non-existent, and thus the shift in image statistics does not account for the observed differences in integration and shape biases (Figure 12).

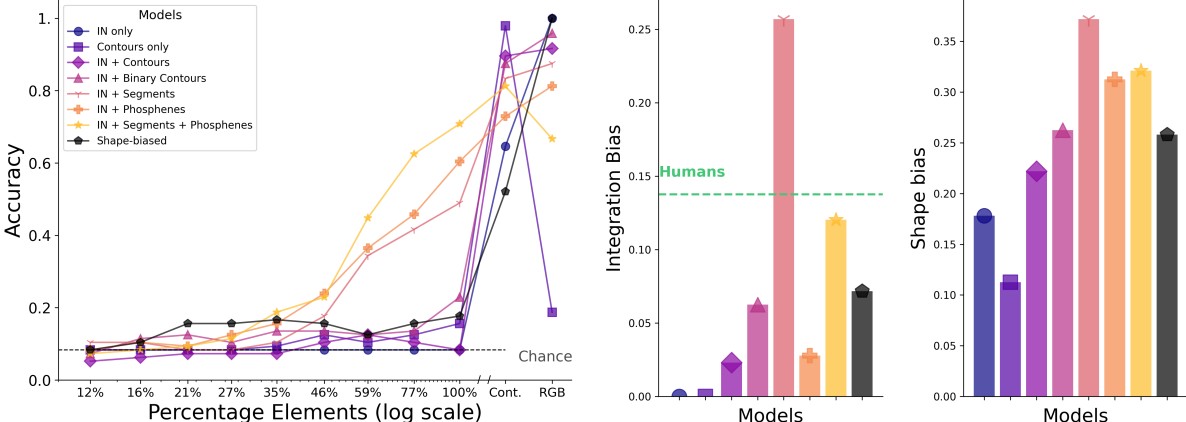

*Figure 12.* **Model performance of models trained on ImageNet-1k and fragmented ImageNets.** All models were evaluated zero-shot. *Left*: Our trained model performance on the fragmented object task. *Middle*: Individual models' integration bias (bars matched in order and color to the left; first two bars show **IN only** and **Contours only**, which are both 0). *Right*: Model shape bias as measured by the cue conflict dataset (Geirhos et al., 2021). Our models trained on phosphenes, segments, and a combination all reach higher shape bias than the Stylized ImageNet-trained model (Geirhos et al., 2018)

## A.7. Further model details

**Pre-trained model selection.** We include some statistics on our model selection. Table 2 includes information about model averages. **N** denotes the number of models or participants analyzed, **Dataset**$_\mu$ denotes the average number of samples in the model architecture family's training dataset. **FLOPs**$_\mu$ denotes the average compute per sample used by the model architecture. **Acc.**$_\mu$ denotes the average performance of models or participants in the group, and **Acc.**$_{max}$ denotes the maximum any individual model of participant achieved across conditions.

The full list of model architecture families we used is: Vision Transformer (Vaswani et al., 2017; Dosovitskiy et al., 2021); ConvNeXt (Liu et al., 2022); ResNet (He et al., 2016); EfficientNet (Tan & Le, 2019); ResNeXt (Xie et al., 2017); MaxViT (Tu et al., 2022); SwinTransformer (Liu et al., 2021); MobileViT (Mehta & Rastegari, 2022); CORnet-S (Kubilius et al., 2019); AlexNet (Krizhevsky et al., 2012); FastViT (Vasu et al., 2023); RegNet (Radosavovic et al., 2020); TinyViT (Wu et al., 2022).

**Model training details for main analyses.** In addition to our pre-trained model set from timm (Wightman, 2019) and taskonomy (Zamir et al., 2018), we trained a set of 514 models from multiple architecture families. The architectures we trained included ResNet-18, 34, 50, 101, 152 (He et al., 2016); ViTT, S, B, L (Dosovitskiy et al., 2021); ConvNeXtT, S, B, L (Liu et al., 2022); EfficientNet-B0, 1, 2 (Tan & Le, 2019); CORnet-S (Kubilius et al., 2019); AlexNet (Krizhevsky et al., 2012).

We trained these datasets using three datasets: ImageNet-1k (Russakovsky et al., 2015), ImageNet-21k (Ridnik et al., 2021), and EcoSet (Mehrer et al., 2021). We trained models on full datasets, as well as subsets of the datasets ranging from ≈500 training samples to the full dataset. For a graphical illustration of all training dataset sizes, see Figure 13. ConvNeXt and ViT models were trained using the original training recipes suggested by the original authors. All other models were trained for 100 epochs using a batch size of 512 with the SGD optimizer with a cosine decay learning rate. The learning rate started at 0.1 and warmed up over 5 epochs, with a weight decay of $10^{-4}$. The horizontal flip and random resized crop augmentations were used.

*Table 2.* Summary of model dataset, compute, and performance on our task. All conditions, including RGB and contours are included. **N** denotes the number of models or participants analyzed, **Dataset**$_\mu$ denotes the average number of samples in the model architecture family's training dataset. **FLOPs**$_\mu$ denotes the average compute per sample used by the model architecture. **Acc.**$_\mu$ denotes the average performance of models or participants in the group, and **Acc.** $_{max}$ denotes the maximum any individual model of participant achieved across conditions.

| Architecture | Zero-shot | | | | | Decoder-fit | | | | |
|---|---|---|---|---|---|---|---|---|---|---|
| | **N** | **FLOPs**$_\mu$ | **Dataset**$_\mu$ | **Acc.**$_\mu$ | **Acc.**$_{max}$ | **N** | **FLOPs**$_\mu$ | **Dataset**$_\mu$ | **Acc.**$_\mu$ | **Acc.** $_{max}$ |
| **Human** | **50** | | | 0.59 | 0.73 | | | | | |
| ViT | 122 | $6.4 \times 10^{10}$ | $1.2 \times 10^8$ | 0.16 | 0.36 | 257 | $1.2 \times 10^{11}$ | $1.7 \times 10^8$ | 0.34 | 0.54 |
| ConvNeXt | 107 | $6.4 \times 10^{10}$ | $1.1 \times 10^8$ | 0.17 | 0.36 | 189 | $6.1 \times 10^{10}$ | $1.2 \times 10^8$ | 0.33 | 0.55 |
| ResNet | 75 | $1.2 \times 10^{10}$ | $3.3 \times 10^7$ | 0.12 | 0.18 | 179 | $1.3 \times 10^{10}$ | $3.3 \times 10^7$ | 0.31 | 0.45 |
| EfficientNet | 45 | $1.6 \times 10^9$ | $7.4 \times 10^6$ | 0.12 | 0.18 | 87 | $1.6 \times 10^9$ | $7.4 \times 10^6$ | 0.29 | 0.39 |
| ResNeXt | 37 | $3.4 \times 10^{10}$ | $8.1 \times 10^7$ | 0.18 | 0.25 | 39 | $3.7 \times 10^{10}$ | $8.4 \times 10^7$ | 0.39 | 0.46 |
| MaxViT | 26 | $1.7 \times 10^{11}$ | $1.3 \times 10^8$ | 0.20 | 0.24 | 30 | $1.6 \times 10^{11}$ | $1.4 \times 10^8$ | 0.40 | 0.46 |
| SwinT | 26 | $5.1 \times 10^{10}$ | $8.0 \times 10^7$ | 0.20 | 0.23 | 34 | $5.4 \times 10^{10}$ | $9.4 \times 10^7$ | 0.43 | 0.48 |
| MobileViT | 16 | $1.0 \times 10^{10}$ | $9.7 \times 10^6$ | 0.17 | 0.19 | 16 | $1.0 \times 10^{10}$ | $9.7 \times 10^6$ | 0.33 | 0.36 |
| CORnet-S | 15 | $3.3 \times 10^{10}$ | $5.3 \times 10^7$ | 0.11 | 0.17 | 30 | $3.3 \times 10^{10}$ | $5.3 \times 10^7$ | 0.29 | 0.37 |
| AlexNet | 15 | $1.4 \times 10^9$ | $6.1 \times 10^7$ | 0.12 | 0.16 | 30 | $1.4 \times 10^9$ | $6.1 \times 10^7$ | 0.30 | 0.33 |
| FastViT | 14 | $6.7 \times 10^9$ | $1.9 \times 10^7$ | 0.19 | 0.21 | 14 | $6.7 \times 10^9$ | $1.9 \times 10^7$ | 0.39 | 0.42 |
| RegNet | 13 | $9.5 \times 10^{10}$ | $1.2 \times 10^8$ | 0.19 | 0.23 | 16 | $9.2 \times 10^{10}$ | $1.3 \times 10^8$ | 0.39 | 0.46 |
| TinyViT | 8 | $1.2 \times 10^{10}$ | $1.5 \times 10^7$ | 0.18 | 0.20 | 11 | $1.0 \times 10^{10}$ | $1.7 \times 10^7$ | 0.41 | 0.45 |
| **Total** | **621** | | | | | **1038** | | | | |

## A.8. Additional details on training datasets.

Our set of models from the timm library (Wightman, 2019) include models trained on various datasets. In addition, we include 23 pre-trained models from the taskonomy library (Zamir et al., 2018). A full list of timm datasets used in our study, in addition to their references, can be found in Table 3. We also include information on the number of models from each training dataset, including our own, which can be found in Figure 13. Models trained on our datasets are a mix of ImageNet-1k, ImageNet-21k, and EcoSet models.

| Dataset | Citation | Size |
|---|---|---|
| ImageNet-1k | (Deng et al., 2009) | 1.28M |
| Ecoset | (Mehrer et al., 2021) | 1.5M |
| ImageNet-12/21k | (Ridnik et al., 2021) | 13.6M |
| YFCC100m | (Thomee et al., 2016) | 100M |
| LVD142M | (Oquab et al., 2023) | 142M |
| LAION-400M | (Schuhmann et al., 2021) | 400M |
| OpenAI | (Radford et al., 2021) | 400M |
| LAION-A | (Schuhmann & Beaumont, 2022) | 939M |
| IG-1B | (Yalniz et al., 2019) | 940M |
| Datacomp | (Gadre et al., 2023) | 1.0B |
| SEER | (Goyal et al., 2021) | 1.0B |
| WebLI-1B | (Chen et al., 2023) | 1.0B |
| 600M-2B | (Schuhmann et al., 2022) | 2.0B |
| DFN2B | (Fang et al., 2023) | 2.0B |
| LAION-2B | (Schuhmann et al., 2022) | 2.2B |
| MetaCLIP | (Xu et al., 2023) | 2.5B |
| SWAG | (Singh et al., 2022) | 3.6B |
| DFN5B | (Fang et al., 2023) | 5.0B |

*Table 3.* Model training datasets used in our study (M = million, B = billion).

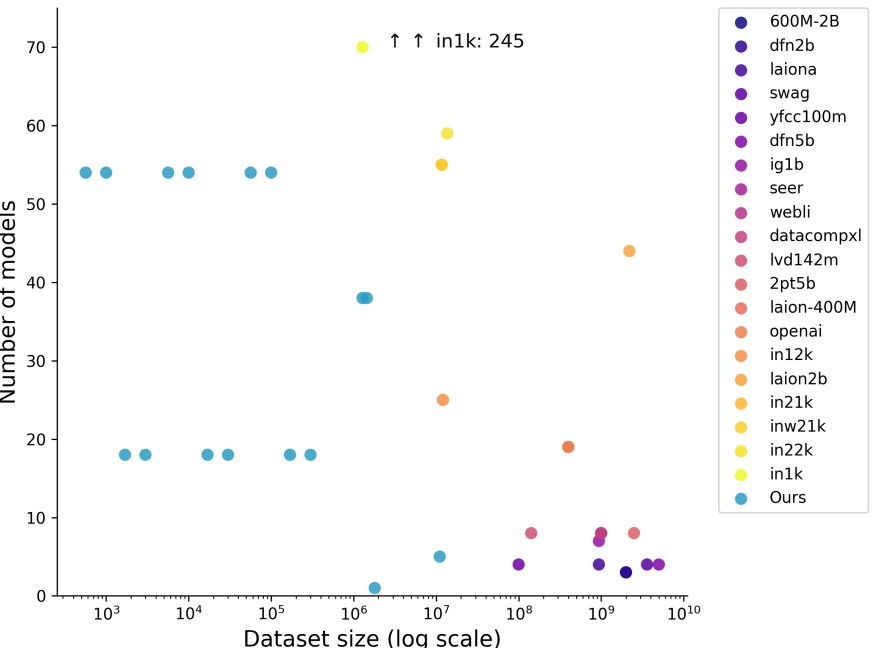

*Figure 13.* Training datasets counts.

### A.9. Training models on fragmented ImageNet

**Fitting Data Preparation.** As a first step, we use the background-removal library of (Gatis, 2023) to remove the background from ImageNet images. Next, we employ the tool from (Rotermund et al., 2024) to convert the background-removed RGB images into contour images. We then use the same tool (Rotermund et al., 2024) to generate two additional transformations: segmented objects and phosphene images. To obtain binary contours, we first employ $3 \times 3$ Gaussian filter and Otsu thresholding on contour images. During this process, a subset of ImageNet images could not be processed. Consequently, we ended up with 1240077 training phosphene/segment/contour images out of the original 1281167.

**Joint Training.** We train multiple instances of ResNet-18 (He et al., 2016) on various combinations of four datasets: *ImageNet*, *ImageNet-contours*, *ImageNet-phosphenes*, and *ImageNet-segments*. We use the standard ResNet-18 architecture without modifications, and we optimize the following multi-task objective:

$$\mathcal{L} \;=\; \lambda_1 \mathcal{L}_{\text{ImageNet}} \;+\; \lambda_2 \mathcal{L}_{\text{contours}} \;+\; \lambda_3 \mathcal{L}_{\text{phosphenes}} \;+\; \lambda_4 \mathcal{L}_{\text{segments}}, \tag{1}$$

where each $\lambda_i$ is set to 1 when the corresponding dataset is included in the training process.

During joint training on multiple datasets, we first select a batch of images and load the variations of those that $\lambda_i \neq 0$. For instance, if the model is trained on base ImageNet and contours, we load a batch of RGB images and their contour counterparts. For phosphene and segment variations, we only use the 100% condition, and load the image accordingly.

Except the ImageNet-trained variant, which is trained using SGD, all model variants are trained with the AdamW (Loshchilov & Hutter, 2019) optimizer using a learning rate of $10^{-3}$, a weight decay of $5 \times 10^{-2}$, and a cosine learning rate decay schedule. We employ a linear warm-up for the first 5 epochs and train for a total of 100 epochs. The batch size is 512, and standard ImageNet augmentations (random resized cropping and horizontal flipping) are applied throughout training.

### A.10. Model set for joint behavioral analysis

In Section 4 we analyzed 50 models on ImageNet-C-top1, and in Appendix A.1 we evaluated the same models on ImageNet-top1. These models were selected primarily based on performance on our fragmented object task. The full procedure for selection was as follows:

- We first chose a small set of common and powerful architecture families: ViT, ConvNeXt, Swin, CvT, EfficientViT,

ConvNeXtv2.

- We then select 20 models from the selected architectures linearly based on their scores on our fragmented object recognition zero-shot task.

- We finally add the best model for any architectures that were not produced above, resulting in a total of 22 models in the first batch.

- We also included in our analysis an additional 28 models trained ourselves from [citation removed for anonymity; relevant details in Appendix A.7], resulting in a total of 50 models analyzed jointly on our task, and common neural and behavioral tasks.

## A.11. Stimulus synthesis

We implemented stimulus synthesis using (Rotermund et al., 2024), which implements the following algorithm to filter image $C(x, y, \varphi)$:

$$(1) \quad G_\vartheta(x, y, \varphi) = \exp\left(-\frac{x^2 + y^2}{2\sigma^2}\right)\left(\cos\left(\frac{2\pi\,(x\cos\varphi + y\sin\varphi)}{\lambda} - \vartheta\right) - c_0\cos\vartheta\right) \tag{2}$$

$$(2) \quad C(x, y, \varphi) = \sqrt{\left[G_{\vartheta=0} * I\right]^2(x, y, \varphi) \; + \; \left[G_{\vartheta=\pi/2} * I\right]^2(x, y, \varphi)} \tag{3}$$

Where

$$c_0 = \exp\left(-2\left(\tfrac{\pi\sigma}{\lambda}\right)^2\right).$$

The convolution operation is implemented via a 2D Fast Fourier Transform, and $I(x, y)$ is the grayscale image pixel value at $(x, y)$. Thus, the resulting contour image $C(x, y, \varphi)$ contains the value of the local contour of orientation $\varphi$ at position $(x, y)$. We set the spatial filter size to $\sigma = 0.06$ arcdeg and the spatial scale to $\lambda = 0.12$ arcdeg.

For the placement of phosphenes and segments, we place elements on the contours of the object preferentially depending on the strength of the contour and its directionality. We use this algorithm both for our experimental stimuli, as well as ImageNet-1k images. For ImageNet-1k images, we also perform a background removal using rembg (Gatis, 2023) before applying this algorithm.

## A.12. GPT-4o experimental methodology

GPT-4o (checkpoint: `gpt-4o-2024-05-13`) was initialized with `temperature=0.0` and presented with a system message:

> You are invited to participate in a research project. The research project aims to study visual perception, i.e. the processing of visual information in participants. In our experiments, visual stimuli will be presented to you as images. Your task will be to evaluate some properties of these stimuli and to give your answer using the provided answer options only. We will measure your behavioral responses (accuracy of your responses, correctness, etc.). Answer as truthfully as possible, and if you are not sure, make your best guess.

Following that, GPT-4o received exactly the same instructions as humans. Experimental procedure followed human experimental procedure exactly. 6 messages was kept in shared message history.

## A.13. Image background color control experiment.

To test whether the specific image color statistics affect model performance or cause difficulties, we tested a small subset of models on red background images. We performed a t-test to test for the difference in the group performance of the fragmented conditions and found that there was no difference in model accuracy under these two conditions ($t = 0.323, p = 0.749$). This shows that the specific model training distribution is not the reason for model performance here, and that the results are robust against a change of background (Figure 15).

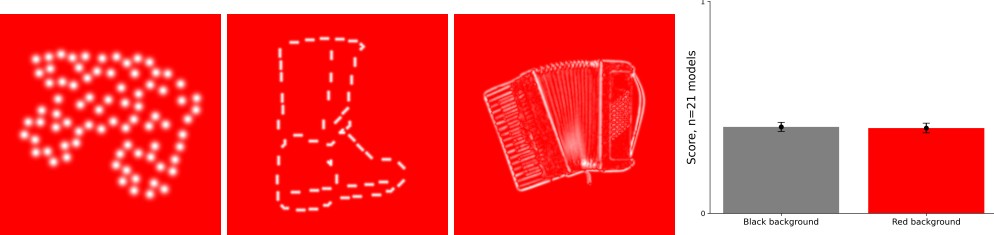

*Figure 14.* **Red background control.** *(Left)*: Examples of images with a red, instead of black, background. *(Right)*: Results show no sensitivity to the background color.

### A.14. Differences between ViT and Convolutional architectures.

We conducted several analyses comparing transformers and CNNs. To make results comparable, we restricted the analysis to only models trained on ImageNet-1k.

We found that the overall performance is no different (CNNs: $32.49\%$ on average, ViTs: $31.24\%$ on average, t-test of means $p = 0.1108$: not significant). Integration biases between transformers and CNNs are comparable: ViT ($7.8\%$) and CNN ($10.1\%$) are not statistically significantly different ($t = 2.45, p = 0.016$) at the 0.01 criterion.

The way model performance scales across the number of elements is the same across models and across conditions: only one condition across all conditions is different between CNNs and ViTs (the 16-phosphenes condition), while the differences between the rest are statistically non-significant.

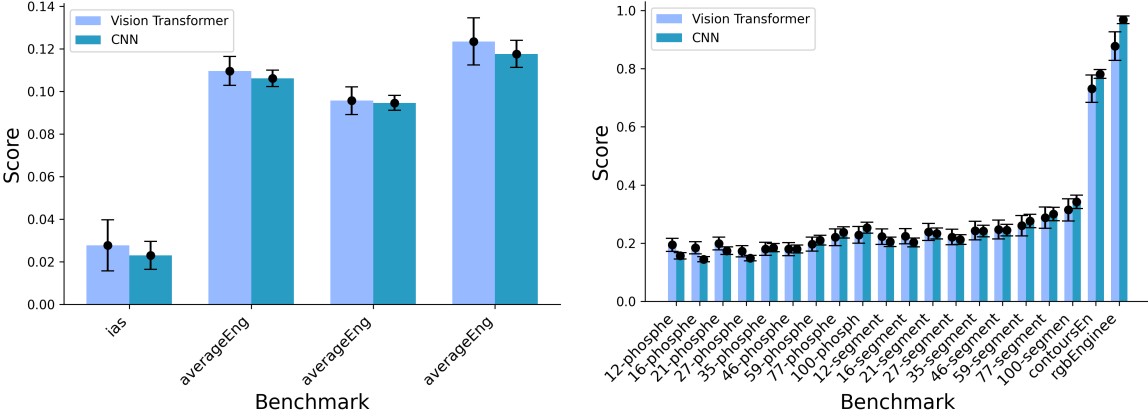

*Figure 15.* **Vit vs CNN analysis.** *(Left)*: Across all models trained on ImageNet, ViTs and CNN do not exhibit differences across any measures. **A**: Integration bias. **B**: Average performance (both conditions). **C**: Average performance (phosphenes). **D**: Average performance (segments). *(Right)*: Performance scaling across percentage of elements.

