# OpenReview forum: "Contour Integration Underlies Human-Like Vision"
_ICML.cc/2025/Conference — ICML 2025 poster_

### Official Review · Reviewer_eQjn · 2025-03-12

**Overall Recommendation:** 3

**Summary:**

The authors systematically dissected where and why models struggle with contour integration by designing an experiment that tested object recognition under various levels of object fragmentation. It was found that humans exhibited an integration bias – a preference towards recognizing objects made up of directional fragments over directionless fragments. It was also found that not only models that shared this property performed better, but that this bias also increased with model training dataset size, and training models to exhibit contour integration leads to high shape bias.

**Claims And Evidence:**

The authors did not explain why the deep learning models performed worse than humans. In vision science, contour integration is normally associated with long-range interactions which has been exploited by humans. However, the authors did not mention this in the paper. The authors may need to investigate the size of the actual receptive field utilized by those CNN models. It is known that CNNs cannot capture long-range interactions/dependencies well while Transformer was developed to address this issue. In this context, it would be interesting to compare CNNs and Transformer networks with regard to the spatial extent that they utilize.

**Essential References Not Discussed:**

Field et al. 1993, Contour integration by the human visual system-- evidence for a local ‘association field’
Panis et al. 2008, Identification of everyday objects on the basis of fragmented outline versions
Dong et al. 2021, Perceptual Texture Similarity Estimation An Evaluation of Computational Features

**Experimental Designs Or Analyses:**

Again, the authors are encouraged to examine the maximal spatial extent that the deep models exploited and the relationship between this value and the performance of the model on contour integration.

**Methods And Evaluation Criteria:**

I would like to see the size of the actual receptive field utilized by those CNN models in the experimental results. It is really interesting to explore the relationship between the performance of those models and the size of receptive field.

**Other Comments Or Suggestions:**

-The difference in human perception between two sets of stimuli should be analyzed.
-The authors are encouraged to investigate the size of the actual receptive field utilized by CNN models.
-CNN and Transformer models should be compared in terms of the spatial extent that they can exploit.

**Other Strengths And Weaknesses:**

Strength:
-A large number of Models were examined.
-The experimental results were analyzed using statistical methods.
-Two sets of stimuli were used.

Weakness:
-The difference in human perception between two sets of stimuli should be analyzed. In my opinion, perception of directionless outlines should have a strong relationship with proximity. However, this is not the case for the directional version.
-Contour integration is normally associated with long-range interactions which can be exploited by humans. But the authors did not mention this in the paper. The authors are encouraged to investigate the size of the actual receptive field utilized by CNN models. It is known that CNNs cannot capture long-range interactions/dependencies well while Transformer was developed to address this issue. Therefore, it would be interesting to compare CNNs and Transformer networks in terms of the spatial extent that they can exploit.

**Questions For Authors:**

None.

**Relation To Broader Scientific Literature:**

The human subject study was similar to that conducted in (Panis et al. 2008). However, the authors did not mention this work at all. Also, they missed many important references, such as (Field et al. 1993), (Dong et al. 2021).

**Theoretical Claims:**

The role of long-range interactions used by HVS should be discussed because they are important to contour integration. The difference in human perception between two sets of stimuli should be analyzed. Note that perception of the outline which consists of directionless points should have a strong relationship with proximity. However, the directional version is different.

---

> ### Author Rebuttal · Authors · 2025-03-31
>
> We thank the reviewer for their review. We have addressed your comments individually below and point to an external link for additional figures: https://drive.google.com/drive/folders/1M_nUONfTXLmUZCHHL0PfIlIaEhpu0vLo?usp=drive_link
>
> > Long-range interactions in humans being normally associated to contour integration, and how that relates to our task.
>
> In our paper, we have intentionally omitted discussion around long-range interactions as our task was not designed to investigate these effects directly:
>
> - Our stimuli spanned only an 8x8 degree window of visual angle.
>
> - Our stimulus presentation time was only 200ms, which was followed by a 1/f noise mask. This ensured that no long-lasting integration or tracing is possible.
>
> The goal of these steps was to ensure that the types of long-range connectivity you refer to are controlled in this setting - see also below for our analysis of the human data.
>
> > The difference in human perception between two sets of stimuli should be analyzed. In my opinion, perception of directionless outlines should have a strong relationship with proximity. However, this is not the case for the directional version.
>
> Thank you for this suggestion, which we have now done. In summary, we found little difference in the scaling of the two conditions (phosphenes vs segments) in humans. The difference in performance in these two cases is mostly a shift of intercept instead of a change in slope across the number of elements (figure __*human_accuracy_split.png*__), though note that while the effect of difference in slopes is small, it is still technically significant (_t=2.87, p<0.01_). This means that the directionality of the segments only slightly affects performance positively over the directionless prosphenes as element density increases.
>
> > Comparison between CNN and transformer architectures, since CNNs do not have long-range interactions while transformers do
>
> Thank you for this suggestion. We indeed included transformers in our original work (the single largest architecture family in our paper is the vision transformer (ViT), of which we had 257 decoder-fit variants; Table 1, row 1).
>
> We have now conducted several new analyses comparing transformers and CNNs. To make results comparable, we restricted the analysis to only models trained on ImageNet-1k, although we also provide figures for all models in the figures too.
>
> - We found that the overall performance is no different (CNNs: _32.49%_ on average, ViTs: _31.24%_ on average, t-test of means _p=0.1108_: not significant). This is also true even when not controlling for dataset (CNNs: _29.08%_ on average, ViTs: _29.98%_ on average, t-test of means _p=0.1889_: not significant); figure __*vit_vs_cnn_imagenet.png*__
>
> - Integration biases between transformers and CNNs are comparable: ViT (_7.8%_) and CNN (_10.1%_) are not statistically significantly different (_t=2.45, p=0.016_) at the 0.01 criterion; figure __*vit_vs_cnn_imagenet.png*__
>
> - The way model performance scales across the number of elements is the same across models and across conditions: only one condition across all conditions is different between CNNs and ViTs (the 16-phosphenes condition), while the differences between the rest are statistically non-significant. __*vit_vs_cnn_imagenet_scaling.png*__
>
> Taken together, these analyses show that long-range interactions do not play a crucial role in our task either in humans nor in models.
>
> > Receptive field sizes and how they relate to task performance should be studied
>
> We thank the reviewer for this nice suggestion. In addition to the experiments where we compared ViTs to CNNs (effectively two extremes of this spectrum), we have now added an explicit test for receptive field size and its relationship to the primate visual system.
> We evaluated the same subset of models as in Fig 6d and 8 on two Brain-Score benchmarks that test for the similarity of the effective receptive field size of a model to a primate counterpart. These are the Grating summation field (GSF) and surround diameter (Marques 2020, Cavanaugh 2002) benchmarks measured in macaque V1. In short, we do not find a statistically significant relationship between either measure of receptive field size similarity and fragmented object recognition accuracy (GSF _r=0.2655, p=0.0653_; surround diameter _r=0.276, p=0.055_); figure __*receptive_field.png*__
>
> We believe these additional results further bolster our conclusion that architecture plays a minimal role in contour integration, and that the receptive field size of models is not a crucial component in this study.
>
> We also thank you for pointing us to missing references, which we have included.
>
> Thank you again for your review. We believe we have addressed all of your concerns and would be grateful if you considered raising your score.

---

> > ### Comment · Reviewer_eQjn · 2025-04-02
> >
> > Many thanks to the authors for addressing my comment. However, I still have some concerns.
> > (1) Within the Abstract, it was stated that "Importantly, humans exhibit an integration bias – a preference towards recognizing objects made up of directional fragments over directionless fragments". This conflicts with the above response,e.g., "In summary, we found little difference in the scaling of the two conditions (phosphenes vs segments) in humans".
> > (2) The receptive field sizes of more models should be investigated. Maybe a CC can be calculated between these sizes and the performances of the models.
> > (3) Since the authors believed that long-range interactions did not take effect in their experiments and there was nothing to do with the size of receptive field, how to explain the phenomena found in the experiments?

---

> > > ### Author Response · Authors · 2025-04-02
> > >
> > > Many thanks for your reply and engagement which we appreciate. We have further comments to make on the concerns:
> > >
> > > **(1)** We want to clarify that when we talk about integration bias, we talk about the difference in performance between the conditions (Figure 6a). In our previous response (which admittedly was left slightly short due to the character limit), we mentioned that the difference in segment and phosphene performance in humans is primarily a **fixed offset** that only changes slightly with the number of elements present in the image (i.e., the _scaling_ of performance in the number of elements is approximately constant). For a figure, see here: https://drive.google.com/file/d/1fNzdf8by5Wlmdve19Drt1_ADS9_YXMBS/view?usp=drive_link
> > >
> > > What this means is that there is a performance difference across all conditions from 12% elements to 100% elements, and this difference in humans is approximately the same on average in all conditions. Thus, our previous comment is not in conflict with our statements about integration bias, but rather show it is stable across conditions.
> > >
> > > **(2)** Since in our previous comment we reported results for exactly this concern with 50 total models, how many models would be satisfactory? This number was chosen similarly to the other reported results in the paper for robustness (Fig 6d) and object recognition (Fig 8), and we would not want to p-hack by adding models until a number becomes significant; but rather commit to a fixed number of models. This number is already rather large given other work in the area (e.g., Dapello, Marques et al. 2020: 30 models; Linsley, et al. 2018: 18 models; Fel, Felipe, Linsley et al. 2023: 84 models, Biscione & Bowers 2023: 16 models) and is consistent with our other analyses.
> > >
> > > Furthermore, if we were to take the current effects of receptive field size similarity that we reported in our previous rebuttal as statistically significant (which they are not): GSF _r=0.2655, p=0.0653_; surround diameter _r=0.276, p=0.055_, the effect sizes would be rather small, with $R^2$ values of _0.070_ and _0.076_ respectively. This is in contrast to training dataset size, which we in the paper reported to have _r=0.814_ with an $R^2$ of _0.663_. Thus, even if one were to test more models and assume that the effect remained the same until the effect was significant, it would still fall massively short of the impact of training dataset size, and short of even the amount of compute a model uses per sample (FLOPs).
> > >
> > > **(3)** We also believe this is a very interesting question, and we have focused on the algorithmic (rather than the mechanistic) explanation in this work, since an investigation of this type has until now been missing. Our current stance on the algorithmic level is that contour integration is a bias that's helpful for solving general tasks - that as the task diversity increases, the model implements contour integration, and that this implementation improves robustness in general.
> > >
> > > This of course does not answer _how_ this contour integration is implemented on a mechanistic level, and perhaps it varies wildly based on model, too. Based on the new results regarding ViT vs CNN, as well as receptive field size experiments, we can say that it's unlikely that receptive field size plays a crucial role, as even if the effects we found were significant, they would be small compared to the total effect of contour integration we report. That being said, we think a full investigation of this would be interesting, but we also firmly believe would warrant its own work due to the extent of investigation, how many different model types, different methods to investigate this, etc., there are.
> > >
> > > Thank you again for your comment. We hope it resolves your concerns and hope that if so, you consider raising your score.
> > >
> > > **References**
> > >
> > > Linsley, D., Kim, J., Veerabadran, V., Windolf, C., & Serre, T. (2018). Learning long-range spatial dependencies with horizontal gated recurrent units. Advances in Neural Information Processing Systems (Vol. 31).
> > >
> > > Fel T, Felipe I, Linsley D, Serre T. Harmonizing the object recognition strategies of deep neural networks with humans. Adv Neural Inf Process Syst. 2022 Dec;35:9432-9446.
> > >
> > > Kubilius, J., Schrimpf, M., Kar, K., Rajalingham, R., Hong, H., Majaj, N., Issa, E., Bashivan, P., Prescott-Roy, J., Schmidt, K., Nayebi, A., Bear, D., Yamins, D. L., & DiCarlo, J. J. (2019). Brain-like object recognition with high-performing shallow recurrent ANNs., Advances in Neural Information Processing Systems (Vol. 32).
> > >
> > > Dapello, J., Marques, T., Schrimpf, M., Geiger, F., Cox, D., & DiCarlo, J. J. (2020). Simulating a primary visual cortex at the front of CNNs improves robustness to image perturbations. Advances in Neural Information Processing Systems (Vol. 33, pp. 13073–13087).
> > >
> > > Biscione, V., Bowers, J.S. Mixed Evidence for Gestalt Grouping in Deep Neural Networks. Comput Brain Behav 6, 438–456 (2023). https://doi.org/10.1007/s42113-023-00169-2

---

### Official Review · Reviewer_RkAR · 2025-03-12

**Overall Recommendation:** 3

**Summary:**

The work investigates the difference between the human ability to generalise object recognition and DNNs. The study builds on experiments that test the ability to recognise objects even in the presence of fragmentation, particularly by contour integration, and the ability of DNNs to perform the same task. The experiments conducted tests with 50 individuals and 1038 models from 13 architecture families and  18 datasets; the largest models were trained on more than 5B images. The tests showed that, in general, people perform better than DNNs and that the performance trend is related to the amount of data, with a correlation of 0.814. On the other hand, architectures are less important than data in contour integration. The set of experiments shows that contour integration is learned automatically from the distribution of data and does not depend directly on horizontal connectivity in the primary visual cortex, as previously assumed, as this mechanism can be learned from the data.

**Claims And Evidence:**

The work follows an interesting track leading to conclusions on contour integration through meaningful experiments on humans in a controlled laboratory setting and on models.

Each experiment step is documented in detail, and metrics and statistics explain the analysis.

**Essential References Not Discussed:**

All required references for the main task seem to be discussed.

**Experimental Designs Or Analyses:**

In my opinion, the experimental design is valid and also quite interesting.

**Methods And Evaluation Criteria:**

Benchmarks are made on a large amount of data collected, both on the side of humans and models, significantly validating the claims.

**Other Comments Or Suggestions:**

I'm unsure about the attempt to train models directly to group elements. Several attempts like this have been made in the literature. Moreover, obtaining binary contour using Gaussian filters and Otsu threshold is quite naive.

Also, apart from  "IN only"  for all the other cases, it is unclear how contours were extracted, as no literature is mentioned.
This ease in extracting contour is a bit superficial since no model performs well on this task yet.
This blurs the conclusions in lines 368-378.

**Other Strengths And Weaknesses:**

The paper is well-written and very interesting.
However, I found that attempting to train models directly to group elements is a bit naive.

There are attempts like this in the literature, and obtaining binary contours using Gaussian filters and Otsu threshold is not quite skillful. Furthermore, besides *IN only*, it is not clear how contours were extracted in the other combination, as there is no reference to the literature. The problem of obtaining contours, which here is considered resolved and easy is, in fact, still an open problem even in the case of supervised models..

**Questions For Authors:**

Please explain clearly how you have extracted the contours on all objects in ImageNet-1K

**Relation To Broader Scientific Literature:**

This is probably the first systematic model on the subject. It will certainly induce discussion and further research; there is little or no prior literature.

**Theoretical Claims:**

Perhaps the only theoretical claim concerns the demonstration that contour integration is not necessarily a product of horizontal connectivity in the primary visual cortex (referring to the literature) and that the mechanism is learned from the amount of data inducing learning. I am not sure that the tests effectively support these conclusions.

---

> ### Author Rebuttal · Authors · 2025-03-27
>
> Thanks for your review. We are happy to hear that you found our experiments meaningful, and the paper well-written and very interesting. We reply point-by-point to your comments below and point to an external link for additional figures: https://drive.google.com/drive/folders/1M_nUONfTXLmUZCHHL0PfIlIaEhpu0vLo?usp=drive_link
>
> > Contour extraction is not explained in the paper
>
> We apologize for not including a detailed description of this step in our paper. We use a phosphene rendering algorithm (Rotermund et al., 2023, as cited in the paper) for extracting contours and rendering stimuli with phosphenes or segments. The contour extraction is simple: images are transformed from RGB to grayscale, and then convolved with a Gabor filter bank of 8 different orientations. Specifically, our contour image is defined by:
>
> __see *contour_equation.png*__
>
> For the placement of phosphenes and segments, we place elements on the contours of the object preferentially depending on the strength of the contour and its directionality.
> We use this algorithm both for our experimental stimuli, as well as ImageNet-1k images. For ImageNet-1k images, we also perform a background removal using `rembg` (Gatis) before applying this algorithm. We will include this description in the updated manuscript.
>
> > The problem of obtaining contours, which here is considered resolved and easy is, in fact, still an open problem even in the case of supervised models
>
> How exactly we obtain contours from RGB images is not central to our argument in any way. This is for two reasons:
>
> 1) all models and humans see the same images, regardless of how the contour was extracted.
>
> 2) The contour condition is merely a control condition meant to show that the large drop in model performance is not merely due to superficial changes in data distribution (i.e., most of the image being black).
>
> The contour extraction merely serves as an image preprocessing step that we can use to render images which are shown to humans and models to compare their performance and behavioral characteristics.
>
> > The paper is well-written and very interesting. However, I found that attempting to train models directly to group elements is a bit naive.
>
> The goal of this experiment was to causally show the primacy of training data in achieving a human-like contour bias, which we successfully demonstrate with the improved performance of the resulting models. Despite the lack of horizontal connectivity or other architectural biases, we were able to train a model simply using segments and phosphenes to exhibit a human-like integration bias. This also resulted in high shape bias, exceeding previous shape biases from other similar direct training approaches (Geirhos et al., 2018). While the approach itself is simple, it serves to strengthen our claim about the importance of dataset.
>
> > There are attempts like this in the literature, and obtaining binary contours using Gaussian filters and Otsu threshold is not quite skillful.
>
> We are not exactly sure what is meant by it not being “quite skillful”. The motivation here was to simply turn our non-binary contours extracted using Gaussian filters into binary contours, and for this purpose the methodology works very well.
>
> > Perhaps the only theoretical claim concerns the demonstration that contour integration is not necessarily a product of horizontal connectivity in the primary visual cortex (referring to the literature) and that the mechanism is learned from the amount of data inducing learning. I am not sure that the tests effectively support these conclusions.
>
> This is a very interesting thought – which experiments do you have in mind that would better support these conclusions? We believe our evidence is strong. We show that:
>
> - human-like contour integration emerges in models trained on large datasets despite any explicit architectural mechanism being present (Figs. 6b, 7). This provides proof of existence that such a mechanism is not strictly necessary.
> - We causally show that human-like contour integration is possible to directly train for without the use of human behavioral response data (Figure 7). This is causal evidence in a controlled setting that such a mechanism is not necessary.
>
> Taken together, we believe these facts strongly support the conclusion that horizontal connectivity is not necessary for contour integration. Of course it is still possible that the human visual system implements contour integration using a mechanism of horizontal connectivity. Our results challenge the prevailing view that this is the only way to implement contour integration, or that the existence of this mechanism is the key reason for why contour integration exists - clearly, it emerges in other settings too.
>
> We thank you again for your review, and believe that we have addressed your comments. In light of this, we hope that you consider raising your score.

---

### Official Review · Reviewer_ToRh · 2025-03-12

**Overall Recommendation:** 3

**Summary:**

This paper conducts a nuanced analysis of the extent to which vision models are human-like by conducting an experiment involving categorization of degraded images, where those images are reduced to lines or to fragments that are either points or line segments. Humans are able to recognize the images with fragmented contours while these pose a challenge for many vision models. Very large models nonetheless approach human performance, although they do not show a bias for directional fragments that is shown in humans.

## update after rebuttal

Thank you for clarifying. These points do not change my opinion of the paper and I will keep my score.

**Claims And Evidence:**

The core claims are justified through careful experiments and statistical analyses. Inferential test statistics and error bars are included.

**Essential References Not Discussed:**

I did not see any major omissions.

**Experimental Designs Or Analyses:**

The basic experimental designs are relatively simple, involving construction of a set of stimuli and evaluating model performance across those stimuli. The set of models used is extensive.

**Methods And Evaluation Criteria:**

In general the methods made sense for this problem and the experiment design was creative and sensible.

**Other Comments Or Suggestions:**

In general the paper was clear and I appreciated the use of color in the text to differentiate models.

**Other Strengths And Weaknesses:**

The primary strengths of this paper are its novel experimental approach and interesting findings about human vision. The main weakness is that it is not clear whether there are actionable insights for improving vision models -- the main focus in the paper in addressing this point is suggesting that there may be omissions from the training data, but this hypothesis is not explored in detail.

**Questions For Authors:**

No questions other than the issues identified above.

**Relation To Broader Scientific Literature:**

There is a fairly extensive literature covering comparisons of models to human performance in image classification tasks. The paper does a good job of summarizing that literature. The key contributions here are use of a novel paradigm to provide a more stringent test of these models and the demonstration that while large models actually approach human performance even in this new task there is still an interesting bias in human vision that differentiates it from the models.

**Theoretical Claims:**

There were no theoretical claims to evaluate as the primary claims are empirical.

---

> ### Author Rebuttal · Authors · 2025-03-31
>
> Thank you for your review. We are glad you find our experiments justified and carefully conducted, and that you found the findings interesting. We respond to each of your comments below point-by-point.
>
> > It is not clear whether there are actionable insights for improving models based on our results
>
> We believe there are quite a number of actionable insights for improving models based on our results:
>
> **First**, many attempts in the literature for reproducing alignment to low-level human visual cortex have focused on architectural changes in lieu of other approaches (such as data-based approaches), see e.g. Kubilius, Schrimpf et al. (2019). We show that these approaches do not pay off when trained and evaluated at scale. This gives a direct actionable insight: for modelers seeking to improve models of basic human vision, a more fruitful approach is to focus on making the training diet of the models more human-like (especially in scale) than to improve the model architecture in a specific way.
>
> **Second**, we demonstrate success in training models for contour integration directly. While this in itself is the less surprising insight, it certainly is valuable for those who simply want models that exhibit human-like contour integration for further experiments. Interestingly, we also find that training for contour integration in models also leads to shape bias.
>
> **Third**, we find that models with a more human-like integration bias exhibit improved accuracy on a downstream object classification task (Figure 6b, Figure 8) – indicating that selecting models for their ability to integrate contextual information is a useful validation signal when building artificial neural networks.
>
> Taken together, our work substantially advances our understanding of contour integration in vision science, and provides a clear path models that do exhibit human-like contour integration: large training diets, or direct training. This finding contrasts previous work that focus on the architectural role of horizontal connectivity in human visual cortex and models as the mechanistic source of contour integration (e.g. Linsley et al, 2018).
>
> > “Very large models nonetheless approach human performance, although they do not show a bias for directional fragments that is shown in humans”
> [...]
> “ [the paper demonstrates] that while large models actually approach human performance even in this new task there is still an interesting bias in human vision that differentiates it from the models”
>
> We would like to clarify that the best models do indeed show a human-like bias (Figure 6b), demonstrating that large training diets can yield human-like contour integration behavior in models. Integration bias separates almost all models from humans due to their integration bias that is not human-like. This shows that object recognition itself is possible without contour integration (a surprising finding in itself: e.g. Field 1993 as pointed out by reviewer eQjn; Kovacs & Julesz 1993; Grossberg & Mingolla 1985), but the largest models learn to do contour integration nonetheless.
>
>
> Thanks again for your review. We believe we have addressed your comments and ask you to consider raising your score?
>
>
>
> **References from all rebuttals**
>
> Cavanaugh, J. R., Bair, W., & Movshon, J. A. (2002). Nature and interaction of signals from the receptive field center and surround in macaque V1 neurons. Journal of Neurophysiology, 88(5), 2530–2546. https://doi.org/10.1152/jn.00692.2001
>
> Zamir, A. R., Sax, A., Shen, W. B., Guibas, L. J., Malik, J., & Savarese, S. (2018). Taskonomy: Disentangling task transfer learning. Proceedings of the IEEE Conference on Computer Vision and Pattern Recognition (CVPR).
>
> Gatis, D. (n.d.). rembg [Computer software]. GitHub. Retrieved March 31, 2025, from https://github.com/danielgatis/rembg​
>
> Linsley, D., Kim, J., Veerabadran, V., Windolf, C., and Serre, T. Learning long-range spatial dependencies with horizontal gated recurrent units. NeurIPS, volume 31. Curran Associates, Inc., 2018.
>
> Grossberg, S., Mingolla, E. Neural dynamics of perceptual grouping: Textures, boundaries, and emergent segmentations. Perception & Psychophysics 38, 141–171 (1985). https://doi.org/10.3758/BF03198851
>
> Kubilius, J., Schrimpf, M., Kar, K., Rajalingham, R., Hong, H., Majaj, N., ... & Dicarlo, J. (2019). Brain-like object recognition with high-performing shallow recurrent ANNs. NeurIPS (pp. 12785-12796).
>
> Field DJ, Hayes A, Hess RF. Contour integration by the human visual system: evidence for a local "association field". Vision Res. 1993 Jan;33(2):173-93. doi: 10.1016/0042-6989(93)90156-q. PMID: 8447091.
>
> Marques, T., Schrimpf, M., & DiCarlo, J. J. (2021). Multi-scale hierarchical neural network models that bridge from single neurons in the primate primary visual cortex to object recognition behavior. bioRxiv. https://doi.org/10.1101/2021.03.01.433495

---

### Official Review · Reviewer_QTFh · 2025-03-15

**Overall Recommendation:** 4

**Summary:**

The paper presents evidence suggesting that contour integration—a fundamental feature of human vision—remains largely absent in artificial vision models. To demonstrate this, the authors tested human performance on contour integration tasks and evaluated over 1,000 computational models to identify trends in machine vision systems. Notably, they found that models trained on larger datasets exhibited better contour integration capabilities. Intriguingly, certain advanced models like GPT-4 achieved performance levels comparable to humans, underscoring the role of scale in bridging this perceptual gap.

**Claims And Evidence:**

I am sympathetic to the argument that models trained on vast datasets may develop shape biases conducive to contour integration. However, given that many state-of-the-art models (e.g., GPT-4o) are trained on proprietary, non-public datasets, there remains ambiguity about whether their training data included images resembling the experimental stimuli. While the paper notes that the exact stimuli used are not publicly disclosed, the possibility of "close encounters" between test stimuli and training data—even unintentional ones—raises concerns about ecological validity. A possible way could be  to verify whether performance persists under novel conditions, such as with modified backgrounds or added noise, which would reduce the likelihood of prior exposure influencing model behavior.

**Essential References Not Discussed:**

I think the paper cites the most important literature.

**Experimental Designs Or Analyses:**

The experiments appear methodologically sound. I specifically reviewed the human data collection protocols and model evaluation framework, which encompass:

Zero-shot evaluation via the BrainScore pipeline.

Decoder fitting using fragmented ImageNet subsets, with label remapping to align with the 12 stimulus classes.

Dataset size analysis to correlate scale with performance.

Architecture size comparisons: While insightful, my primary concern lies in the potential unfairness of comparing models like RNNs—which lack access to modern large-scale training datasets—against newer architectures (e.g., transformers) trained on vastly larger corpora. This discrepancy in data availability complicates direct performance comparisons, as differences may reflect dataset scale rather than architectural superiority, leading to t values that undermined possible architecture impact.

Experiments about integration biases and how countour integration leads to robustness.

**Methods And Evaluation Criteria:**

Yes.

**Other Comments Or Suggestions:**

No.

**Other Strengths And Weaknesses:**

The paper is well writen and provides good justification for the different experimental choices. It is transparent about its limitations and provide some possible way to mitigate some of the problems that can be raised.   I think it is a good example of a task that can help understanding the gap between human and artificial vision.

Maybe I missed it, but perhaps would be to comment on models that have been trained on other downstream tasks such as object detection, image segmentation, and how the hypothesis of data diets apply or not in them.

**Questions For Authors:**

*  Although its mention in the limitation section, I think the potential problem of data leackage in the larger datasets can be an important point for the validity of the study. It would be good to run at least one control, perhaps under different background colors or under some noise.

**Relation To Broader Scientific Literature:**

The paper delivers valuable insights for the community by systematically evaluating how artificial vision models align with human visual processing. It  contextualizes its findings against prior work, contrasting architectural approaches (e.g., Linsley et al.’s biologically constrained networks) with data-driven explanations for emergent capabilities. Specifically, the authors highlight how integration bias—a phenomenon extensively studied in models by Geirhos et al.—can arise from large-scale training, thereby enabling certain architectures to solve contour integration tasks without explicit architectural mimicry of human vision. This dual focus on architecture and data provides a nuanced framework for understanding the mechanisms behind model performance, guiding future research toward more human-like machine vision systems.

**Theoretical Claims:**

No proofs were provided in the paper.

---

> ### Author Rebuttal · Authors · 2025-03-31
>
> Thank you for your favorable review. We are happy to read that you found the paper insightful and our experiments sound, and that you see how it helps guide future research. We address your comments point by point below, and point to an external link for additional figures: https://drive.google.com/drive/folders/1M_nUONfTXLmUZCHHL0PfIlIaEhpu0vLo?usp=drive_link
>
> > Concerns about training data, which could be remedied by testing models on the task with e.g. a different background that is more unlikely to be in training datasets
>
> Thank you for the nice suggestion. We followed your proposal and made a new version of our dataset, this time with a red background instead of the standard black background (image examples in the folder __*red_images/*__). We tested the subset of pre-trained models also in figures 6d and figure 8. We did a t-test to test for the difference in the group performance of the fragmented conditions and found that there was no difference in model accuracy under these two conditions (_t=0.323, p=0.749_). This shows that the specific model training distribution is not the reason for model performance here, and that the results are robust against a change of background. See figure __*normal_vs_red.png*__.
>
> > Concerns about the analysis regarding how important architecture is and how robust the analysis is
>
> We have taken two additional steps to remedy this:
> We now also run a mixed effects regression, which allows for the architecture type to be treated as a random variable. This analysis also finds that training dataset size is most important (_z=14.66, p<0.001_), while model compute is less important but significant (_z=7.177, p<0.001_). Importantly, the random effect for architecture type under this more controlled analysis is dominated by error rather than systematic variance (variance=0.002, standard error = 0.008), suggesting that the random factor of architecture family plays little systematic role in the mixed effects model.
> We analyzed ViT vs CNN-based architectures in more detail for a fixed training dataset size (ImageNet-1k) -- due to space constraints, we detail this analysis in our response to reviewer __eQjn__.
> These further analyses provide additional evidence to our claim that architecture is less important than training dataset size.
>
> > Different downstream tasks, such as segmentation
>
> We thank the reviewer for this idea. We analyzed additional models from the taskonomy (Zamir et al. 2018) model set, which  provides a great degree of breadth in terms of tasks, with the downside that we only have one model per task. Nevertheless, we report these additional results here (Figure __*taskonomy_scores.png*__).
>
> In summary:
> - The base object classification and scene classification models reach 31% & 32% accuracy respectively
> - Unsupervised segmentation reaches 29%%, surface normal estimation 33%, and depth estimation reach 34% accuracy
> - Interestingly, the edge-computing model reaches only 21% accuracy
>
> In general, it seems that while the task itself appears to play a role, object recognition stands out as a good objective, with more specialized objectives often falling slightly short. Nevertheless, the differences are rather small.
>
> > Novelty of the work: “Specifically, the authors highlight how integration bias—a phenomenon extensively studied in models by Geirhos et al.—can arise from large-scale training [...]”
>
> We would like to point out that Geirhos et al. (2018) did **not** study integration bias, but rather shape bias, which is a different metric measured using an entirely different dataset. Specifically, Geirhos et al. (2018) studied the human and model responses to images with a conflicting texture and shape cue. This gives a point estimate of the amount of shape bias a model has.
>
> Our work does not study shape bias, but rather contour integration. Shape bias is a human preference to detect object categories by their shape, rather than their texture, while contour integration is the ability to integrate elements despite their discontinuity in image space. As such, while contour integration allows the recognition of images that would otherwise be unrecognisable, shape bias is a description of choice preferences. Our experimental setup is also different. Instead of several different conditions that are not related to each other on a single continuous scale (like in Geirhos et al.), we varied the image fragmentation. This allowed us to understand where and why neural networks fail (at the fragmentation of the contour), and to pinpoint this specifically to contour integration, a well-established “algorithm” in humans. Taken together, our work is very different from previous work, with a novel experiment, metric, and findings.
>
> We thank you again for your review. We believe these comments address your concerns, but please let us know if additional analyses would be helpful. If all concerns are addressed, we ask the reviewer to consider increasing their score.

---

### Decision · Program_Chairs · 2025-05-01

**Decision:**

Accept (poster)

**Comment:**

This work introduced some ideas from cognitive psychology to improve visual recognition models. Focusing on the idea of contour integration, the proposed additions to the models lead to a subtle improvement in recognition accuracy. There is overall minimal but positive enthusiasm for the contributions of this work.